# Assessment of the Diet Quality Index and Its Constituents in Preschool Children Diagnosed with a Food Allergy as Part of the “Living with an Allergy” Project

**DOI:** 10.3390/nu17101724

**Published:** 2025-05-20

**Authors:** Malgorzata Kostecka, Julianna Kostecka, Paulina Kawecka, Magdalena Sawic

**Affiliations:** 1Department of Chemistry, Faculty of Food Science and Biotechnology, University of Life Sciences, Akademicka 15, 20-950 Lublin, Poland; paulina.kawecka@up.lublin.pl; 2Faculty of Medicine, Medical University of Lublin, Chodźki 19, 20-093 Lublin, Poland; kostecka.julianna@gmail.com; 3Student Scientific Society of Dietitians, Faculty of Food Science and Biotechnology, University of Life Sciences, Akademicka 15, 20-950 Lublin, Poland; magdalenasawic@icloud.com

**Keywords:** food allergy, elimination diet, Healthy Diet Index, unhealthy dietary patterns, nutrition knowledge

## Abstract

Pediatric food allergies (FAs) are health conditions that adversely impact the quality of life of children and their caregivers. Aim: The primary objective of the present study was to assess the quality of the diets administered to allergic children based on the Healthy Diet Index (HID-10), to determine the influence of parental knowledge about FAs and the elimination diet, and to identify the factors that contribute to healthy food choices. Material and Methods: This study was conducted as part of the “Living with an Allergy” research and educational program for preschool children, which was implemented between June 2021 and June 2023 in the city of Lublin. Results: Food allergies were diagnosed and confirmed in 241 children, including 106 boys (44%). A higher number of unhealthy dietary factors (DQI-1) was significantly associated with gender, and lower DQI values were more often noted in boys (*p* < 0.05). In turn, a higher number of health-promoting dietary factors (DQI-3) was significantly associated with a younger age in children (OR 1.54; 95%CI 1.17–1.74, *p* < 0.01) and with an older age in parents (OR 1.43; 95%CI 1.2–1.67, *p* < 0.05). Conclusions: Children whose diets, including the necessary modifications, were recommended by a physician or a dietitian were characterized by significantly higher DQI values and a higher number of health-promoting dietary factors. The diets of children with FAs should consist mainly of unprocessed foods to control the intake of unhealthy products that suppress immunity.

## 1. Introduction

The early years of life are important for growth and development and for shaping healthy behaviors in the future. An optimal diet during childhood may have a protective effect against negative health outcomes such as obesity. The intake of selected foods and beverages in children and adolescents has attracted research interest in Poland and other European countries. The Polish authorities have implemented several programs aiming to shape healthy eating habits among youngsters [1,2] and adopted a resolution on dietary guidelines for school meals and foods that may be sold in schools and other educational institutions [3].

Dietary indices are used in assessments of the quality of children’s diets to monitor children’s eating habits and behaviors and to predict the risk of lifestyle diseases in adulthood [4]. Dietary indices quantify the subject’s overall adherence to dietary guidelines and patterns that are recommended for specific age groups in different countries. These indices are developed with the use of traditional dietary assessment methods such as food-frequency questionnaires or simple questionnaires based on national dietary guidelines and recommendations [5,6]. However, dietary assessments in children have many limitations. Research has shown that children younger than 8 years are unable to recall the consumed foods, accurately describe portion size, or develop conceptual frameworks for correctly assessing the time and frequency of food consumption [7]. For this reason, research examining young children’s diets must be conducted with the parents’ or caregivers’ involvement [5].

Pediatric food allergies (FAs) are health conditions that adversely impact the quality of life of children and their caregivers in the United States and the European Union, including Poland [8,9]. Meal planning for children with FAs affects the quality of life, and food intolerance invariably necessitates changes in a family’s eating habits. Elimination diets associated with allergies to specific food allergens can pose an additional challenge to healthy nutrition and balanced meal planning in the pediatric population [10]. Children with FAs need to avoid exposure to allergy triggers, and they need to follow an elimination diet that differs from the diet consumed by their peers, for as long as deemed necessary by the physician. The extent to which these dietary restrictions affect dietary diversity, healthy eating behaviors, and diet quality has not been fully elucidated [11,12]. The influence of parental factors, socioeconomic factors, and parental knowledge about FAs and dietary restrictions on children’s eating habits and adherence to an elimination diet also remains unclear [13]. Research has demonstrated that children adhering to an elimination diet have a reduced quality of life, especially in the absence of dietary counselling [12]. Based on recent evidence, the quality of the elimination diet has emerged as an increasingly important concern, and the main focus has shifted from the quantity of food consumed to the quality and selection of wholesome foods [14]. These changes indicate that the quality of the diet consumed by children with FAs should be accurately assessed using tools appropriate for the pediatric population [15].

The impact of FAs and the elimination diet on the quality of Polish children’s diets has not been investigated to date. Therefore, the primary objective of the present study was to assess the quality of the diets administered to allergic preschoolers based on the Healthy Diet Index (HID-10), to determine the influence of parental knowledge about FAs and the elimination diet, the type of food allergy, and the child’s and parents’ age on diet quality, and to identify the factors that contribute to healthy food choices. The secondary objective was to identify meal planning factors that contribute to unhealthy dietary patterns.

## 2. Materials and Methods

### 2.1. Study Design and Participants

This study was conducted as part of the “Living with an Allergy” research and educational program for preschool children, which was implemented between June 2021 and June 2023 in the city of Lublin in south-eastern Poland. Ethics approval was by the Ethics Committee of the Medical University of Lublin (KE-0254/273/2021, 2021-04-21). A total of 2760 parents of preschoolers (aged 3–6 years) attending 37 public kindergartens and 11 private kindergartens were invited to participate in educational workshops organized as part of the above program. The following exclusion criteria were applied in the present study: absence of a diagnosed allergy, observance of a different elimination diet, such as a gluten-free diet, or a disease requiring a specialized diet, including diabetes, galactosemia, or phenylketonuria. Children with feeding disorders were also excluded from this study. The inclusion criteria were a medical diagnosis of a food allergy and the absence of metabolic disorders requiring a different restrictive diet. Based on these criteria, 241 parents whose children had been diagnosed with a food allergy (not later than 6 months prior to this study) were qualified for the described part of this study. The research tool was a survey questionnaire which was applied to examine the child’s diet and frequency of consumption of different foods, and to assess the prevalence of allergies and the parents’ nutritional knowledge regarding the safety of an elimination diet. The questionnaires were developed specifically for the project, but they were based on validated and published questionnaires (Infant and Young Child Feeding (IYCF) Assessment, KomPAN^®^, SF-FFQ4PolishChildren^®^). All of the applied questionnaires had been previously assessed in a control group and used in previous studies [16,17,18].

All participants gave their voluntary consent to participate in the survey, were informed about the purpose of the study, and were assured that the study was anonymous. The questionnaires were completed independently by the mothers without any assistance from the researchers. The questionnaires were distributed by the researchers and university students during meetings in kindergartens, and by kindergarten teachers to preschooler groups. The parents took 25 min on average to complete the questionnaire administered in the kindergarten, and the questionnaires to be filled at home were returned after 10 days on average.

The questionnaire consisted of three parts. The first part was a qualitative survey, and the respondents rated the frequency with which their children consumed various groups of food products on a scale, where the responses ranged from “never” to “several times a day”. The questionnaire included questions on the frequency of consumption of 30 food categories. The results were processed using daily consumption frequency indicators (times per day) to ensure consistent interpretation of the data (Table 1) [19,20].

The second part of the questionnaire was designed to elicit information about the diagnosis of the food allergy, type of allergy, type of allergen, duration and stages of the elimination diet, types of eliminated food products, and difficulty in following an elimination diet assessed subjectively by the parents on a scale of 0–20 points, where 0 points denoted the absence of difficulties, and 20 points denoted extreme difficulty. The respondents were also asked about the frequency of consultations with a dietitian and an allergy specialist, medical recommendations, and the observed changes in the family’s diet and eating habits.

Similarly to the authors’ previous study [17], the third part of the questionnaire consisted of 15 questions on parental knowledge, and the respondents could choose one of three answers: true, false, or I don’t know. These questions assessed parental knowledge about allergens, elimination diets, cross-reactivity in allergic reactions, and food-allergen labeling. The respondents received 1 point for a correct answer and 0 points for an incorrect answer or “I don’t know” answer. Points were summed up for each respondent (range: 0 to 15 points). Based on tertile distribution, the respondents were divided into three nutrition knowledge categories: bottom (0–7 points), middle (8–11 points), and upper tertile (12–15 points).

The questionnaire also contained demographic questions about the child’s age and gender, and the parent’s age, education, and place of residence.

### 2.2. Diet Quality Indices

Based on methodological recommendations [19,20], the quality of the diet was assessed with the use of three indicators:Healthy Diet Index (HDI-10, Healthy-Diet-Index-10) covering 10 food groups with potentially beneficial health effects;Unhealthy Diet Index (UDI-12, Unhealthy-Diet-Index-12) covering 12 food groups with potentially adverse health effects;Diet Quality Index (DQI, Diet-Quality-Index) covering 22 food groups, including 10 food groups with potentially beneficial health effects and 12 food groups with potentially adverse health effects.

Diet quality scores are so-called hypothesis-driven dietary patterns, i.e., they describe a set of commonly occurring dietary characteristics that have been selected on the basis of available scientific evidence. Diet quality scores can be calculated and interpreted alternatively or simultaneously with dietary patterns that have been identified from data analysis using advanced statistical methods [19].

The above indices were calculated by summing up the frequency of consumption (times per day) of 10 food groups covered by HDI-10, 12 food groups covered by UDI-12, and 22 food groups covered by the DQI (Table 2 and Table 3).

The Healthy Diet Index, Unhealthy Diet Index, and Diet Quality Index were calculated for each individual respondent according to the formulas given below. All calculations were based on the daily consumption frequency (times per day) shown in Table 1. The results obtained were then presented as an average for the study population and for study characteristics such as gender and parental knowledge.

The summed frequency of consumption (times/day) was converted and expressed on a scale of 0 to 100 points [19] to standardize the scope of indices HDI and UDI and to facilitate their interpretation:Healthy Diet Index (HDI, points) = (100/20) × consumption frequency of 10 food groups (times per day)Unhealthy Diet Index (UDI, points) = (100/24) × consumption frequency of 12 food groups (times per day)

The Diet Quality Index was calculated by summing up all HDI constituents with a positive sign and all UDI constituents with a negative sign. Weighting indices (weights) were used in the calculations to equate the contribution of the 10 components of the HDI index to the contribution of the 12 components of the UDI index. The DQI ranged from −100 to 100 points [19].Diet Quality Index (DQI, points) = (100/20) × consumption frequency of 10 food groups (times per day) + (−100/24) × consumption frequency of 12 food groups (times per day)

### 2.3. Data Analysis

Categorical variables were presented as sample percentages (%), and continuous variables were expressed by median values and the interquartile range (IQR). The differences between groups were analyzed in the chi-squared test (categorical variables) or the Mann–Whitney test (continuous variables). The Kruskal–Wallis test was applied to analyze the relationships between variables in more than two mutually independent groups. Before statistical analysis, data were checked for normal distribution in the Kolmogorov–Smirnov test. The significance level was set at *p* < 0.05.

Categorical variables were analyzed using logistic regression models. The odds ratios (OR) and 95%CI were calculated. The significance of OR was verified by the Wald test. The following confounders were included in the logistic regression analysis: gender, age (years), parent’s gender, parent’s age (years), DQI, nutritional knowledge score, elimination diet, consultation with a dietitian, and use of substitutes in the elimination diet. In each analysis, a set of confounders was selected based on the modeled research question. The results of all tests were regarded as statistically significant at *p* < 0.05. Data were processed in the Statistica program (version 13.1 PL; StatSoft Inc., Tulsa, OK, USA; StatSoft, Krakow, Poland).

## 3. Results

Food allergies were diagnosed and confirmed in 241 children, including 106 boys (44%). General information about the examined subjects and the diagnosed allergies is presented in Table 4.

The child’s age at the onset of the first symptoms was significantly correlated with the type of allergen. An allergy to cow’s milk proteins was diagnosed at the earliest age; the mean age at diagnosis was 2.6 ±1.4 months, and gender was not a differentiating factor (*p* > 0.05). The second earliest diagnosed allergy was the allergy to small-seeded fruits, and the mean age at diagnosis was 7.1 ± 2.3 months. In turn, the allergy to chicken egg whites was diagnosed relatively late (past the age of 1 year) in all children, and the mean age at diagnosis was 1.4 ± 0.3 years in girls and 1.7 ± 0.4 years in boys (*p* < 0.05). According to the surveyed parents, allergies to food additives, mainly food colorants, were diagnosed latest at the mean age of 2.3 ± 0.6 years, irrespective of the child’s gender. Children aged 3–4 years attended regular consultations with an allergy specialist more often than older children (*p* < 0.05), children diagnosed with multiple FAs (*p* < 0.05), and children whose both parents were allergic (*p* < 0.05).

The diet quality indices calculated based on the consumption frequency of selected food groups are presented in Table 5. Boys scored more points in UDI-12, and fewer points in HDI-10 and DQI, than girls.

The diets of the studied children were characterized by a low number of health-promoting factors and unhealthy factors. The average value of the DQI for all participants was 4.47 points, and it was significantly differentiated by gender (*p* < 0.05) and the level of parental knowledge (*p* < 0.05) (Table 6).

A detailed analysis demonstrated that in the group of food products, meals, and beverages with potentially beneficial health effects, fruit were consumed most frequently, significantly more frequently by girls (OR 1.35, 95%CI 1.1–1.53, *p* < 0.05) and children whose elimination diet was recommended by a pediatrician and on which a dietitian was consulted (OR 1.39; 95%CI 1.06–1.49, *p* < 0.05); vegetables were consumed more frequently by girls (OR 1.54, 95%CI 1.23–1.7, *p* < 0.01), younger children (OR 1.29; 95%CI 1.12–1.36, *p* < 0.05), and children whose diets were characterized by a high number of health-promoting factors (DQI-3) (OR 1.56; 95%CI 1.27–1.77, *p* < 0.01). Milk and milk substitutes (in an elimination diet) were consumed frequently by all children regardless of their gender and age, more frequently by children whose parents were classified in the upper tertile based on their nutritional knowledge score (OR 1.43; 95%CI 1.19–1.57, *p* < 0.01) and children of parents older than 35 years (OR 1.29; 95%CI 1.05–1.38, *p* < 0.05).

In the group of food products and beverages with potentially adverse health effects, foods made of refined flour were consumed most frequently, significantly more frequently by children whose parents were classified in the bottom tertile based on their nutritional knowledge score (OR 1.36; 95%CI 1.16–1.52, *p* < 0.05) and children whose diets were characterized by a high number of unhealthy dietary factors (DQI-1) (OR 1.35; 95%CI 1.09–1.46, *p* < 0.05).

The analysis revealed that adherence to the DQI-2 model was the only factor that was not dependent on gender. In the remaining models, gender was a differentiating factor for the entire study population (Table 7).

A detailed analysis demonstrated that a higher number of health-promoting dietary factors (DQI-3) was significantly associated with a younger age in children (OR 1.54; 95%CI 1.17–1.74, *p* < 0.01) and with an older age in parents (OR 1.43; 95%CI 1.2–1.67, *p* < 0.05). Children whose diets, including the necessary modifications, were recommended by a physician and on which a dietitian was consulted were characterized by significantly higher DQI values. The incorporation of safe food substitutes to elimination diets to minimize nutritional deficiencies was also associated with a higher number of health-promoting dietary factors (OR 1.76; 95%CI 1.22–1.89, *p* < 0.01), more often in girls (OR 1.41, 95%CI 1.17–1.56, *p* < 0.05) and in parents with a high level of nutritional knowledge (OR 1.77; 95%CI 1.43–2.03; *p* < 0.01). The results are presented in Table 8.

Potentially allergenic foods need to be identified and eliminated from a child’s diet to effectively manage an FA. The difficulty in adhering to an elimination diet was related not only with parental knowledge regarding the role of the diet in managing a food allergy, but also with the main allergen (Figure 1). Only around 60% of the surveyed parents were able to correctly identify the dietary sources of potentially allergenic cow’s milk proteins, especially in processed foods, and this percentage was significantly higher among older parents (>35 years) with university education and a history of allergy in the family (*p* < 0.05). All parents were familiar with dietary sources of fish, citrus fruit, and chicken egg allergens. The surveyed parents found it most difficult to identify the dietary sources of nuts and food additives as potentially allergenic food ingredients. Only one in four parents correctly identified breakfast products (cereal, muesli, bread spreads), sweets (candy bars, chocolate), and ready-made milk-based desserts as potential sources of nuts. Parents aged < 35 years with education other than university education gave the highest number of incorrect answers (*p* < 0.05), whereas 34% of the surveyed parents were unable to identify synthetic colors and the corresponding labelling on food packaging. Only 21% of the parents, mostly those with university education (*p* < 0.05), correctly identified potentially allergenic foods containing soy protein.

The greatest difficulties in adhering to an elimination diet were noted in children allergic to nuts and food additives, where HDI and UDI were within the moderate range of values in both cases. The highest HDI values were observed in the elimination diets of children allergic to cow’s milk proteins and egg whites, and difficulties in following these diets were low. The highest UDI values were noted in the elimination diets of children allergic to fish proteins and citrus fruit, whereas the lowest UDI values were observed in children allergic to cow’s milk proteins and food additives (Figure 2 and Figure 3).

## 4. Discussion

Pediatric allergic diseases have emerged as one of the most common chronic conditions affecting children in developed countries. The diagnosis always poses a challenge for the family and requires changes in the diet and meal composition, and a knowledge of food allergens. Despite the increase in the prevalence of pediatric FAs, the access to reliable knowledge has not increased, which can lead to many developmental disorders in young children adhering to a long-term elimination diet.

An elimination diet can be challenging for both the child and the parents [21]. In the present study, allergens that are commonly found in various types of foods or result from cross-contamination during the production process (such as nuts, soy, or food additives) posed the greatest obstacle to following a safe elimination diet. Parents whose children were diagnosed with an allergy to food additives found it most difficult to eliminate these allergens from the child’s diet. Food additives can elicit the same responses as sensitizing food products, and the mechanisms responsible for the harmful effects of these substances have not been fully elucidated. According to research, food additives can trigger or increase the severity of allergic disorders or non-allergic hypersensitivities (atopic dermatitis, contact dermatitis, hives, asthma), or exhibit toxic effects [22,23]. The parents of children with an allergy to nuts, including peanuts, should carefully read food labels to assess the product’s safety. In this study, the parents of more than 1/3 of the preschoolers allergic to nuts were uncertain whether they eliminated all potentially sensitizing foods from their children’s diets. According to the surveyed parents, these difficulties resulted mainly from misleading nutrition labels. Patients with an allergy to nuts are often uncertain whether they should eliminate all nuts or only clinically significant nuts from their diet [24]. Avoidance of all nuts seems to be the simplest decision because it minimizes the risk of an immediate response caused by accidental exposure or misjudgment. However, this approach can lead to the unnecessary elimination of nuts that are safe, which can influence the nutritional quality of the patient’s diet [25,26]. Zuberbier et al. suggested that 0.5 mg of protein per 100 g of processed food should be the threshold for voluntary declaration of food allergen traces in processed food [27]. Global differences in allergen labeling may also lead to accidental exposure [28]. Patients should carefully read nutritional and allergen labels on packaged foods. Food allergen apps can also be helpful in interpreting labeling information [28,29], and nutritional counselors should teach allergic patients and their families how to read and interpret food labels.

Many research studies have demonstrated that in children with FAs, an elimination diet can negatively affect their food energy intake and consumption of macronutrients, in particular protein and essential fatty acids, as well as micronutrients, especially calcium and vitamin D [30,31]. The Diet Quality Index (DQI) can be helpful in estimating the consumption frequency of foods with potentially beneficial and adverse health effects. The DQI-I has been applied in a limited number of studies to assess the quality of the diets consumed by healthy children and adolescents [32,33], but very few such studies were conducted on children with FAs. Kalmpourtzidou (2021) found that the avoidance of specific foods or food groups may increase the risk of low diet quality in children with FAs, but low diet quality appears to be a problem in the general pediatric population [12]. In the study population, regular consultations with a dietitian and an elimination diet recommended by a physician were associated with a higher number of health-promoting dietary factors (DQI-3). According to the majority of dietary guidelines for children with FAs [34], individual consultations with a dietitian [14] are the preferred option to ensure a child’s optimal growth and nutritional status [25]. The authors’ previous study revealed that the parents of children diagnosed with an FA and allergenic cross-reactivity should always consult a dietitian, and that online support groups play an important role, but cannot replace specialist consultations [17]. An elimination diet should target specific allergy symptoms and the clinical diagnosis based on the patient’s detailed medical history and an interpretation of test results [25,35]. The patient’s nutritional requirements, the availability of safe food substitutes, and the family’s cooking and meal preparation skills should be considered in comprehensive dietary recommendations [36].

Research studies investigating the mechanisms underlying the relationship between the consumption of processed foods rich in simple sugars and saturated facts and the development of allergies have shown that ultra-processed foods (UPFs) reduce the counts of beneficial bacteria such as *Bacteroidetes* and *Faecalibacterium prausnitzii* and promote colonization by harmful and pro-inflammatory bacteria, thus altering the composition of the gut microbiome. In some studies, intestinal dysbiosis was a typical feature in children diagnosed with allergic diseases [37,38,39].

Unprocessed and minimally processed foods should be prioritized over UPFs in nutritional management of FAs because this approach prevents the development of other chronic diseases. Despite the general scarcity of research on UPF consumption by children with FAs, studies have shown that children and adolescents with FAs consume large quantities of UPFs, which increases their intake of simple sugars and saturated fats [36]. In the current study, the respondents consumed more food products with potentially adverse health effects than those with potentially beneficial health effects. On average, fast foods were consumed once a week by children aged 3–6 years and up to several times per week by boys consuming sweets. According to the latest European Academy of Allergy and Child Immunology (EAACI) Task Force report, UPF consumption can also increase the risk of allergic diseases in healthy children [39]. In children, the consumption of fruit juice, sugar-sweetened beverages, high-carbohydrate UPFs, and snacks containing glucose–fructose syrup, added flavor enhancers, and glycation end products (AGEs) has been associated with allergic diseases. In many but not all studies, exposure to UPFs appears to be associated with an increased prevalence of allergic diseases, such as asthma, wheezing, food allergies, atopic dermatitis, and allergic rhinitis [39]. Similar observations were made in a Hungarian cross-sectional study [40], which revealed an association between asthma and higher intake of fast foods and beverages containing additives. Elias et al. [41] found that active asthma was associated with UPF intake in the past 7 days in Brazilian adolescents. However, not all results are conclusive, and the Pelotas birth cohort study [42] conducted on 6-year-olds demonstrated that UPF intake was not significantly associated with asthma diagnosis or wheezing at 11 years of age.

Ultra-processed foods have poor nutritional composition because they are rich in saturated fats, salt, and sugar, but deficient in vitamins, minerals, and fiber [40,43]. Consequently, UPF consumption may contribute to low nutrient intake and lower diet quality, including during childhood.

Research has shown that a high level of parental knowledge about health and nutrition was associated with better health outcomes and more desirable eating behaviors in children with FAs [17]. However, the parents’ knowledge and attitudes towards nutrition and the elimination diet have been rarely studied [44]. In the present study, a high level of parental knowledge was significantly associated with a lower number of unhealthy dietary factors and fewer problems in adhering to an elimination diet. Other researchers also found that individuals with FAs and individuals from families with a history of FAs had much higher allergy knowledge scores than persons who had no experience with an FA or an elimination diet [45,46]. Taha et al. also found that a higher knowledge score was significantly associated with parenting a child with an FA, higher education and income, the number of discussions held with a healthcare professional regarding FAs, and the preference to acquire information from a dietitian and a physician [44].

### Strengths and Limitations

The strength of this study was a large sample size that was representative of the population of parents of allergic preschoolers, which supported a reliable determination of the prevalence of FAs in kindergarten children. The use of validated questionnaires, in particular the food frequency questionnaire, enabled the assessment of diet quality and the calculation of the Healthy Diet Index (HDI-10) and the Unhealthy Diet Index (UDI-12). This is the first study to calculate and interpret the Diet Quality Index (DQI) in the Polish population of children with FAs. The associations between different types of food allergens, the parents’ nutritional knowledge, and the DQI should be explored to improve the quality of elimination diets in the pediatric population and to develop targeted educational programs for parents, caregivers, and the personnel of educational institutions, focusing on the principles of formulating safe and properly balanced elimination diets. The limitation of this study was the fact that only qualitative methods were used to evaluate the quality of preschoolers’ diets and consumption frequency of different food products. In line with the methodology, the results obtained were used to assess the overall quality of the diet. Quantitative methods such as 3-day food records or 24 h recalls were not applied. The present findings, particularly the values of UDI-12, may provide a basis for future quantitative research to assess the actual intake of saturated fatty acids, simple sugars (including fructose), dietary fiber, and daily energy balance in the context of chronic disease prevention in later life.

## 5. Conclusions

Elimination diets administered to children with FAs are challenging for both the child and the entire family. Problems with formulating and following an elimination diet are reported by all parents of allergic children, but the degree of difficulty in adhering to an elimination diet is closely related to the parents’ nutritional knowledge and the type of allergen.

The DQI and its constituents provide accurate information about the types of consumed foods and the number of health-promoting and unhealthy dietary factors. Patients with FAs need to restrict their intake of various food groups and replace sensitizing products with safe alternatives, and the results of studies examining unhealthy dietary factors can be used in health education programs.

The diet of children with FAs should consist mainly of unprocessed foods, and the intake of unhealthy products should be controlled to improve the quality of the diet and the children’s nutritional status, and to reduce the severity of allergic symptoms in children who are allergic to food additives.

The results of this study indicate that children with FAs should regularly attend consultations with a dietitian to improve the quality of the diet and expand the parents’/guardians’ knowledge on the safe use of elimination diets.

## Figures and Tables

**Figure 1 nutrients-17-01724-f001:**
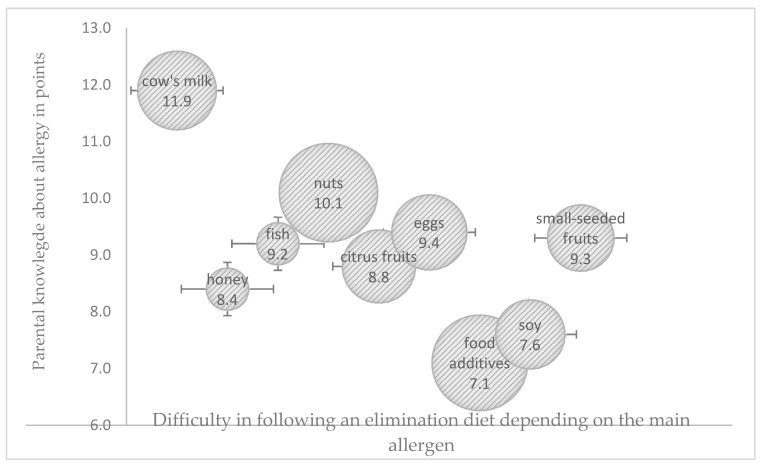
Influence of parental knowledge score (in points) on difficulties in following an elimination diet, depending on the main allergen (the size of the bubble corresponds to the degree of difficulty in following the elimination diet as perceived by the parents).

**Figure 2 nutrients-17-01724-f002:**
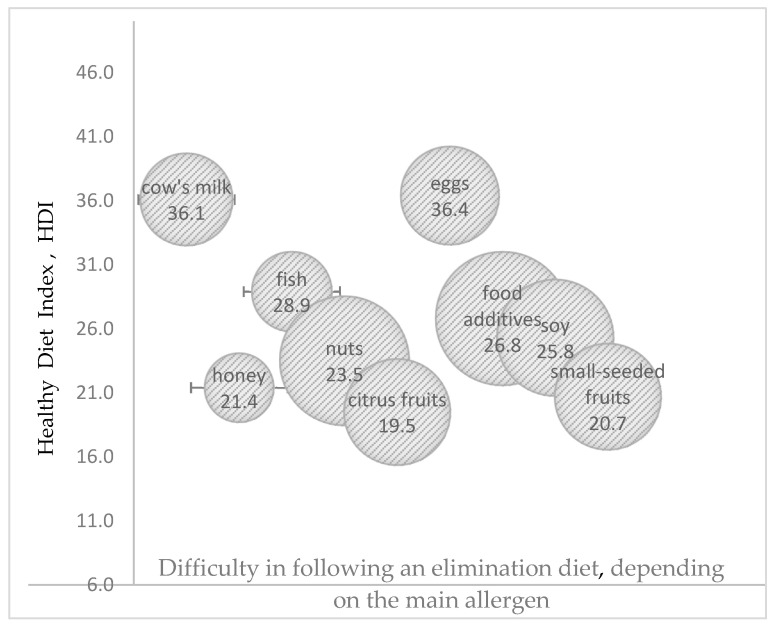
The effect of difficulty in adhering to an elimination diet, depending on the main allergen (the size of the bubble corresponds to the degree of difficulty in adhering to an elimination diet as perceived by the parents), on the value of the Healthy Diet Index (HDI) (in points).

**Figure 3 nutrients-17-01724-f003:**
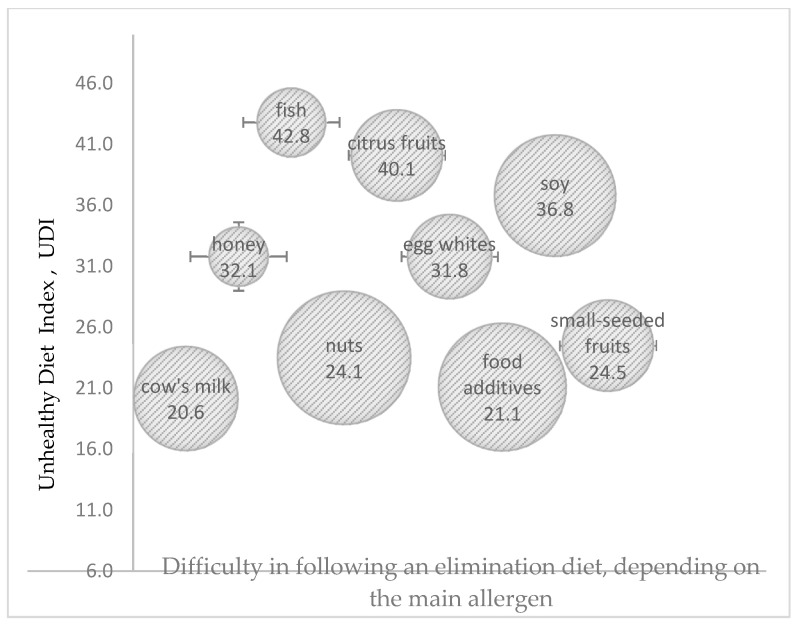
The effect of difficulty in adhering to an elimination diet, depending on the main allergen (the size of the bubble corresponds to the degree of difficulty in adhering to an elimination diet as perceived by the parents), on the value of the Unhealthy Diet Index (UDI) (in points).

**Table 1 nutrients-17-01724-t001:** Consumption frequency indicators.

Consumption Frequency	Ranks Assigned to Consumption Frequency	Daily Consumption Frequency (Times per Day)
Never	1	0
1–3 times per month	2	0.06
Once a week	3	0.14
Several times a week	4	0.5
Once daily	5	1
Several times a day	6	2

**Table 2 nutrients-17-01724-t002:** Constituents of the Healthy Diet Index (HDI) covering food groups with potentially beneficial health effects.

Question	Food Groups Covered by the Healthy Diet Index (HDI)
23	Wholemeal bread
25	Buckwheat groats, oatmeal, wholemeal pasta, and other coarse cereals
31	Milk (including flavored milk, cocoa, milk-based coffee) or milk substitutes in an elimination diet
32	Fermented milks such as yogurt and kefir (plain or flavored)
33	Cottage cheese (tvorog, cream cheese, cheese-based desserts)
37	White meat (e.g., chicken, turkey, rabbit)
38	Fish
40	Legumes (e.g., beans, peas, soybeans, lentils)
42	Fruit
43	Vegetables
HDI = total consumption frequency of 10 food groups (times per day; range: 0–20)	

**Table 3 nutrients-17-01724-t003:** Constitutes of the Unhealthy Diet Index (UDI) covering food groups with potentially adverse health effects.

Question	Food Groups Covered by the Unhealthy Diet Index (UDI)
22	White bread (e.g., wheat, rye, mixed rye and wheat bread, toast bread, bread rolls, crescent rolls)
24	White rice, plain pasta, or fine groats such as semolina and couscous
26	Fast foods (e.g., French fries, hamburgers, pizza, hot-dogs, toasted baguettes)
27	Fried meat and fried dough
28	Butter used as a bread spread or added to cooked, fried, or baked foods
29	Lard used as a bread spread or added to cooked, fried, or baked foods
34	Yellow cheese (including processed cheese spreads and soft-ripened cheese)
35	Cold meats and sausage
36	Red meat (e.g., pork, beef, veal, mutton, lamb, venison)
44	Sweets (candies, cookies, cakes, chocolate bars, muesli bars, and other confectioneries)
46	Canned meat
51	Sweetened carbonated and non-carbonated soft drinks (e.g., Coca-Cola, Pepsi, Sprite, Fanta, orange drinks, lemonade)
UDI = total consumption frequency of 12 food groups (times per day; range: 0–24)	

**Table 4 nutrients-17-01724-t004:** Demographic data of children with diagnosed allergies.

Parameter	Total	Gender		*p*
		Male n [%]	Female n [%]	
Age				
3–4 years	122	57 [46.7%]	65 [53.3%]	<0.05
5–6 years	119	49 [41.2%]	70 [58.8%]	<0.05
Age at diagnosis (in months)	6.7 ± 3.3	6.1 ± 2.9	7.2 ± 3.1	>0.05
Type of allergen *				
Food allergy, including:	241	106 [43.9%]	135 [56.1%]	<0.05
Cow’s-milk protein	137	61 [44.5%]	76 [55.5%]	<0.05
Chicken egg white	39	14 [36.9%]	25 [64.1%]	<0.05
Soy	33	16 [48.5%]	17 [51.5%]	>0.05
Small-seeded fruits	33	17 [51.5%]	16 [48.5%]	>0.05
Citrus fruits	31	19 [61.3%]	12 [38.7%]	<0.05
Nuts	25	9 [36%]	16 [64%]	<0.05
Cacao	15	7 [46.7%]	8 [53.3%]	>0.05
Honey	12	4 [33.3%]	8 [66.7%]	<0.05
Food dyes	10	7 [70%]	3 [30%]	<0.05
Celery	6	4 [66.7%]	2 [33.3%]	<0.05
Fish	6	3 [50%]	3 [50%]	>0.05
Allergies in the family				
None	63	29 [46%]	34 [54%]	<0.05
One parent	71	34 [47.8%]	37 [52.2%]	>0.05
Both parents	53	25 [47.2%]	28 [52.8%]	>0.05
Other family members	54	18 [33.3%]	36 [66.7%]	<0.05
Healthcare/dietary consultations *				
Regular consultations with a dietitian	36	14 [38.9%]	22 [61.1%]	<0.05
Sporadic consultations with a dietitian	46	26 [56.5%]	20 [43.5%]	<0.05
No consultations with a dietitian	159	66 [41.5%]	93 [58.5%]	<0.05
Regular consultations with an allergy specialist	89	39 [43.8]	50 [56.2%]	<0.05
No consultations with an allergy specialist	152	67 [44.1%]	85 [55.9%]	<0.05

* Multiple answers possible.

**Table 5 nutrients-17-01724-t005:** Diet quality in the study sample.

Diet Quality Indices	Total (n = 241)		Girls (n = 135)		Boys (n = 106)		*p*-Value (Mann–Whitney U Test)
	M ± SD	Min–Max	M ± SD	Min–Max	M ± SD	Min–Max	
HDI-10	35.7 ± 11.6	11.6–58.3	37.6 ± 9.5	15.2–58.3	33.9 ± 8.3	11.6–53.1	<0.001
UDI-12	31.23 ± 10.1	8.91–62.5	29.5 ± 8.9	8.91–47.5	34.7 ± 11.7	11.3–62.5	<0.001
DQI	4.47 ± 11.3	−26.6–39.6	8.1 ± 11.6	−21.4–38.6	−0.8 ± 9.15	−26.6–39.6	<0.001

UDI-12, Unhealthy Diet Index (range: 0 to 100 points); HDI-10, Healthy Diet Index (range: 0 to 100 points); DQI, Diet Quality Index (range: −100 to 100 points).

**Table 6 nutrients-17-01724-t006:** Diet Quality Index (DQI) values in relation to child sex and parental nutrition knowledge.

Diet Quality Index (DQI) and Its Constituents	Total	Girls	Boys	*p*-Value in M-W U Test	Parental Knowledge			
					Bottom Tertile (0–7 Points)	Middle Tertile (8–11 Points)	Upper Tertile (12–15 Points)	*p*-Value
Constituents with a Positive Sign								
Wholemeal bread	0.48 ± 0.26	0.51 ± 0.29	0.43 ± 0.21	<0.05	0.31 ± 0.17	0.45 ± 0.3	0.67± 0.49	<0.05
Buckwheat groats, oatmeal, wholemeal pasta, and other coarse cereals	0.36 ± 0.22	0.37 ± 0.25	0.33 ± 0.17	Ns	0.16 ± 0.05	0.33 ± 0.21	0.39 ± 0.24	Ns
Milk (including flavored milk, cocoa) or milk substitutes in an elimination diet	0.88 ± 0.5	0.85 ± 0.44	0.94 ± 0.41	Ns	0.89 ± 0.39	0.66 ± 0.36	0.97 ± 0.56	<0.05
Fermented milks such as yogurt and kefir (plain or flavored)	0.8 ± 0.36	0.91 ± 0.58	0.73 ± 0.36	<0.05	0.8 ± 0.28	0.76 ± 0.48	0.99 ± 0.61	<0.05
Cottage cheese (tvorog, cream cheese, cheese-based desserts)	0.76 ± 0.52	0.7 ± 0.4	0.81 ± 0.32	<0.05	0.75 ± 0.36	0.71 ± 0.36	0.88 ± 0.42	Ns
White meat (e.g., chicken, turkey, rabbit)	0.97 ± 0.51	0.82 ± 0.39	1.03 ± 0.62	<0.05	0.93 ± 0.47	0.90 ± 0.49	0.99 ± 0.54	Ns
Fish	0.14 ± 0.1	0.17 ± 0.12	0.11 ± 0.05	Ns	0.1 ± 0.03	0.14 ± 0.06	0.2 ± 0.05	Ns
Legumes (e.g., beans, peas, soybeans, lentils)	0.22 ± 0.1	0.11 ± 0.08	0.29 ± 0.1	<0.05	0.11 ± 0.03	0.17 ± 0.07	0.26 ± 0.08	<0.05
Fruit	1.53 ± 0.88	1.66 ± 0.92	1.32 ± 0.88	<0.05	1.17 ± 0.57	1.39 ± 0.45	1.93 ± 0.71	<0.05
Vegetables	1.0 ± 0.56	1.42 ± 0.71	0.82 ± 0.5	<0.05	0.56 ± 0.34	1.04 ± 0.59	1.56 ± 0.77	<0.05
Total consumption frequency of 10 food groups—constituents with a positive sign (times per day)	7.14 ± 3.11	7.52 ± 3.14	6.78 ± 2.98	<0.05	6.59 ± 2.87	7.03 ± 3.06	7.71 ± 3.29	<0.05
Constituents with a negative sign								
White bread (e.g., wheat, rye, mixed rye and wheat bread, toast bread, bread rolls, crescent rolls)	1.43 ± 0.71	1.42 ± 0.77	1.45 ± 0.7	Ns	1.67 ± 0.78	1.51 ± 0.51	1.11 ± 0.56	<0.05
White rice, plain pasta, or fine groats such as semolina and couscous	0.89 ± 0.39	0.88 ± 0.45	0.92 ± 0.52	Ns	0.97 ± 0.44	0.93 ± 0.31	0.67 ± 0.34	<0.05
Fast foods (e.g., French fries, hamburgers, pizza, hot-dogs, toasted baguettes)	0.34 ± 0.3	0.26 ± 0.18	0.42 ± 0.15	<0.05	0.41 ± 0.27	0.36 ± 0.19	0.2 ± 0.11	<0.05
Fried meat and fried dough	0.67 ± 0.34	0.56 ± 0.33	0.94 ± 0.51	<0.05	0.71 ± 0.24	0.69 ± 0.36	0.57 ± 0.23	Ns
Butter used as a bread spread or added to cooked, fried, or baked foods	0.73 ± 0.27	0.72 ± 0.29	0.74 ± 0.27	Ns	0.70 ± 0.19	0.77 ± 0.23	0.74 ± 0.26	Ns
Lard used as a bread spread or added to cooked, fried, or baked foods	0.15 ± 0.08	0.15 ± 0.06	0.15 ± 0.05	Ns	0.16 ± 0.05	0.15 ± 0.04	0.14 ± 0.05	Ns
Yellow cheese (including processed cheese spreads and soft-ripened cheese)	0.96 ± 0.59	1.03 ± 0.52	0.91 ± 0.44	Ns	0.91 ± 0.51	1.03 ± 0.66	0.71 ± 0.37	<0.05
Cold meats and sausage	0.67 ± 0.39	0.57 ± 0.36	0.88 ± 0.39	<0.05	0.63 ± 0.36	0.7 ± 0.38	0.59 ± 0.29	Ns
Red meat (e.g., pork, beef, veal, mutton, lamb, venison)	0.32 ± 0.11	0.28 ± 0.19	0.44 ± 0.2	<0.05	0.14 ± 0.07	0.31 ± 0.17	0.46 ± 0.31	<0.05
Sweets (candies, cookies, cakes, chocolate bars, muesli bars, and other confectioneries)	0.72 ± 0.27	0.54 ± 0.37	0.89 ± 0.51	<0.05	0.96 ± 0.39	0.5 ± 0.37	1.04 ± 0.59	<0.05
Canned meat	0.07 ± 0.04	0.07 ± 0.04	0.07 ± 0.04	Ns	0.07 ± 0.04	0.07 ± 0.04	0.07 ± 0.04	Ns
Sweetened carbonated and non-carbonated soft drinks (e.g., Coca-Cola, Pepsi, Sprite, Fanta, orange drinks, lemonade)	0.54 ± 0.3	0.59 ± 0.27	0.510 ± 0.23	Ns	0.66 ± 0.32	0.54 ± 0.37	0.23 ± 0.16	<0.05
Total consumption frequency of 12 food groups—constituents with a negative sign (times per day)	7.49 ± 3.17	7.07 ± 2.98	8.32 ± 3.17	<0.05	7.99	7.56	6.53	<0.05
Diet Quality Index (points)	4.47 ± 11.3	8.1 ± 11.6	−0.8 ± 9.15	<0.05	0.37 ± 8.11	3.63 ± 10.3	11.22 ± 9.14	<0.05

**Table 7 nutrients-17-01724-t007:** Percentage of respondents in each interval of the Diet Quality Index.

Number of Dietary Factors	Total n [%]	Girls n [%]	Boys n [%]	*p*-Value
High number of unhealthy dietary factors—DQI-1	9 [3.7%]	3 [33.3%]	6 [66.7%]	<0.05
Low number of unhealthy dietary factors and health-promoting dietary factors—DQI-2	187	98 [52.4%]	89 [47.6%]	Ns
High number of health-promoting dietary factors—DQI-3	45 [18.7%]	34 [75.5%]	11 [24.5%]	<0.05

**Table 8 nutrients-17-01724-t008:** Odds ratios (95% confidence interval) in an analysis of the relationships between DQI and social factors (children and parents) and the management of an elimination diet.

	DQI-1	DQI-2	DQI-3
Child’s age (ref. 5–6 years): 3–4 years	0.92 (0.88–1.04)	1.18 (1.03–1.3)	1.54 ** (1.17–1.74)
Child’s gender (ref. girls): boys	1.29 * (1.04–1.38)	1.07 (0.93–1.16)	0.76 * (0.59–0.87)
Allergies in the family (ref. no allergies in the family)	1.11 (0.94–1.26)	1.08 (0.88–1.28)	1.17 (1.03–1.31)
Parental education: primary education (ref. university education)	1.39 * (1.08–1.54)	1.31 * (1.12–1.59)	0.74 * (0.61–0.92)
Parental age: >35 Years (ref. < 35 years)	1.03 (0.94–1.2)	1.12 (1.03–1.26)	1.43 * (1.2–1.67)
Parental knowledge: parents in the bottom tertile (ref. parents in the upper tertile)	1.56 ** (1.21–1.85)	1.09 (0.94–1.27)	0.88 (0.76–1.03)
The elimination diet was recommended by a pediatrician and a dietitian was consulted on it (ref. the elimination diet was recommended by a pediatrician, but a dietitian was not consulted on it).	0.69 * (0.57–0.87)	1.06 (0.95–1.17)	1.39 * (1.15–1.53)
Allergenic foods were replaced with safe substitutes to minimize the loss of nutrients (ref. safe substitutes were not used).	0.76 * (0.63–0.92)	1.11 (0.89–1.21)	1.76 ** (1.22–1.89)

Statistically significant: * *p* < 0.05; ** *p* < 0.01;

## Data Availability

Due to ethical restrictions and participant confidentiality, data cannot be made publicly available.

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
