# Peer review of "Assessment of the Diet Quality Index and Its Constituents in Preschool Children Diagnosed with a Food Allergy as Part of the “Living with an Allergy” Project"

_nutrients, 2025, doi:10.3390/nu17101724_

Round 1

Reviewer 1 Report

Comments and Suggestions for Authors

General comments:

Although the study objective is of interest, the methodology appears weak, the statistical analysis lacks clarity, and the results are overall difficult to interpret.

Methods:

The authors state that a validated questionnaire was modified to assess consumption frequencies. However, the specific modifications are not reported. It is essential to clearly describe these changes to determine whether they are minor (and thus compatible with the original tool) or substantial enough to require revalidation.

The frequency categories are quite broad (e.g., “several times a week” could range from 2 to 6 times), which introduces significant imprecision in estimating consumption. Moreover, the absence of quantitative data on the amounts consumed further increases uncertainty in assessing dietary intake.

In the original studies, were the dietary indices based solely on consumption frequencies? I attempted to consult reference 18, but noticed that there are two references listed as number 18, and neither corresponds to the review cited in the text. Furthermore, what is the rationale for rescaling these indices to a 100-point scale rather than using the original scoring system?

It is also unclear what questions were used to evaluate knowledge of food allergies. It does not appear that a validated questionnaire was used, which greatly limits the reliability of this outcome.

Statistical analysis:

The statistical analysis section lacks clarity. The authors state that continuous variables are reported as median and IQR, yet the results also include means and standard deviations. Furthermore, non-parametric tests were used, but it is not clear whether the data met the assumptions of normality. This raises concerns about the appropriateness of the statistical tests employed.

There is no explanation of how the odds ratios (ORs) were calculated. Additionally, it is unclear how the predictors were included in what seems to be a logistic regression model: were they entered all at once or individually? What is the dependent variable? Which are the main predictors? Were potential confounders considered? This section requires thorough revision.

Results:

The results are overly detailed and confusing. It is recommended to streamline this section, reporting only the most relevant findings that align with the study objectives.

Lines 208–224 refer to ORs supposedly derived from Table 6, which only presents means and standard deviations, without clear indication of how the ORs were obtained.

A similar issue arises in line 234, where ORs are reported and linked to Table 7, which only presents percentages. This makes it difficult to follow the reasoning and assess the validity of the conclusions.

Author Response

Thank you very much for all your valuable comments which will help to improve our manuscript. We hope that the corrections and clarifications made will enhance the value of the manuscript. All changes have been marked in red.

Comment 1: The authors state that a validated questionnaire was modified to assess consumption frequencies. However, the specific modifications are not reported. It is essential to clearly describe these changes to determine whether they are minor (and thus compatible with the original tool) or substantial enough to require revalidation.

Response 1: Thank you for this comment. We would like to explain that according to the recommendations of the KomPAN questionnaire development team [Wadolowska et al 2013], the researcher is free to choose questions that are most aligned with the purpose of the study and the researcher’s interests. For the needs of the present study, the questionnaire was shortened and divided into three thematically distinct parts, as described in the Methodology. The introduced modifications were described in detail previously [Kostecka et al., 2022].

Comment 2: The frequency categories are quite broad (e.g., “several times a week” could range from 2 to 6 times), which introduces significant imprecision in estimating consumption. Moreover, the absence of quantitative data on the amounts consumed further increases uncertainty in assessing dietary intake.

Response 2: In accordance with the developed methodology, the amounts consumed were expressed in qualitative terms, and the questions are “question-scales”, where the frequency of consumption was ranked in an increasing order, from “never” to “several times a day”. As shown in Table 1, the ranks and/or daily frequency indices were expressed as “times/day” to standardize the elaboration and interpretation of the results. The proposed ranking and the numerical values assigned to each consumption frequency are consistent with the methodology for interpreting the results of the KomPAN questionnaire [Wadolowska et al 2013] and have been used in many publications. The relevant explanation was provided in the Methods section.

Comment 3: In the original studies, were the dietary indices based solely on consumption frequencies? I attempted to consult reference 18, but noticed that there are two references listed as number 18, and neither corresponds to the review cited in the text. Furthermore, what is the rationale for rescaling these indices to a 100-point scale rather than using the original scoring system?

Response 3: Thank you for bringing our attention to this error. The methodology for the present study was described in detail by Wadolowska et al. (2013). The relevant references were corrected.

It should also be noted that diet quality scores are referred to as “hypothesis-driven dietary patterns” that describe common dietary characteristics selected based on the available scientific evidence. Diet quality indices can be calculated and interpreted separately or together with dietary patterns. Dietary indices were calculated by summing the frequency of intake (times/day) of 10 healthy food groups (HDI), 12 potentially unhealthy food groups (UDI), and all 22 food groups (DQI).  According to the authors of this method, researchers can modify the structure of the indices based on their knowledge and scientific evidence.
As recommended by Wadolowska et al. (2013), the overall intake frequency (times/day) should be recalculated and expressed on a scale of 0 to 100 points to standardize the scope of indices pHDI and nHDI and facilitate their interpretation. The relevant explanation was provided in the Methods section.

Comment 4:  It is also unclear what questions were used to evaluate knowledge of food allergies. It does not appear that a validated questionnaire was used, which greatly limits the reliability of this outcome.

Response 4: The revised manuscript contains a reference to the authors’ previous study which contains a detailed description of the questions assessing parental knowledge about allergies, elimination diets, and sources of allergens.

Comment 5: The statistical analysis section lacks clarity. The authors state that continuous variables are reported as median and IQR, yet the results also include means and standard deviations. Furthermore, non-parametric tests were used, but it is not clear whether the data met the assumptions of normality. This raises concerns about the appropriateness of the statistical tests employed.

There is no explanation of how the odds ratios (ORs) were calculated. Additionally, it is unclear how the predictors were included in what seems to be a logistic regression model: were they entered all at once or individually? What is the dependent variable? Which are the main predictors? Were potential confounders considered? This section requires thorough revision.

Response 5:  Means with a 95% confidence interval (CI) and the percentage distribution of participant characteristics were calculated. The significance of ORs was verified by the Wald test. The normality of distribution of continuous variables in the entire study population was checked by the Kolmogorov–Smirnov test. The description of the methodology was completed and revised.

Comment 6: The results are overly detailed and confusing. It is recommended to streamline this section, reporting only the most relevant findings that align with the study objectives.

Response 6: Thank you for this comment. We agree that some descriptive results were redundant. Our intention was to present the results in a broad context. However, we agree with the Reviewer's suggestion, and the description of the results was shortened.

Comment 7: Lines 208–224 refer to ORs supposedly derived from Table 6, which only presents means and standard deviations, without clear indication of how the ORs were obtained.

A similar issue arises in line 234, where ORs are reported and linked to Table 7, which only presents percentages. This makes it difficult to follow the reasoning and assess the validity of the conclusions.

Response 7: Thank you for this comment. The described results were not linked to the correct tables. The description of the data in Tables 6 and 7 was changed, with a clear indication of how the ORs were obtained.

Reviewer 2 Report

Comments and Suggestions for Authors

The study by Malgorzata Kostecka et al. assessed the quality of the diets administered to allergic children to determine the influence of parental knowledge about FA and the elimination diet and to identify the factors that contribute to healthy food choices. The study is informative and interesting. I have the following questions and comments:

1, the exclusion criteria of the study should also be provided in line 89. 

2, the mechanisms underlying the development of  food allergies and intake of processed food should be discussed. 

3, in line 396-397, the authors concluded that "The diets of children with FA should consist mainly of unprocessed foods to control the intake of unhealthy products that suppress immunity." Could unhealthy products suppress immunity or activate immunity in children with FA ? In the present study, the authors have not checked the changes of the immunity parameters. I think the conclusions should be revised.

4, in all the figures, the authors should specify the error bars. Is it SD or SEM? 

5, line 398 to line 400, I think this is not a conclusion. This is more like a suggestion. 

Author Response

Thank you very much for all your valuable comments which will help to improve our manuscript. We hope that the corrections and clarifications made will enhance the value of the manuscript. All changes have been marked in green.

Comment 1: The exclusion criteria of the study should also be provided in line 89. 

Response 1: Thank you for this comment. The exclusion criteria were provided.

Comment 2: The mechanisms underlying the development of  food allergies and intake of processed food should be discussed. 

Response 2: Potential links between the consumption of processed foods and the mechanisms underlying allergy development were discussed in a paragraph dedicated to ultra-processed foods.

Comment 3: In line 396-397, the authors concluded that "The diets of children with FA should consist mainly of unprocessed foods to control the intake of unhealthy products that suppress immunity." Could unhealthy products suppress immunity or activate immunity in children with FA ? In the present study, the authors have not checked the changes of the immunity parameters. I think the conclusions should be revised.

Response 3: Thank you for this comment. Indeed, the study did not evaluate the effects of UPFs on the immune system, and the statement was not based on the results of the study, but on a review of the literature. The indicated sentence was revised.

Comment 4: In all the figures, the authors should specify the error bars. Is it SD or SEM? 

Response 4: The values of the error bars were not indicated in the figures because the only purpose of the drawings was to compare the effect of difficulty in adhering to an elimination diet (as perceived by the parents) on the values of both dietary indices. The size of the bubble denotes the average number of responses (on a scale of 0-20) concerning specific allergens.

Comment 5: Line 398 to line 400, I think this is not a conclusion. This is more like a suggestion. 

Response 5: We agree with the Reviewer. The sentence was revised accordingly.

Round 2

Reviewer 2 Report

Comments and Suggestions for Authors

The authors have addressed my major concerns. It can be considered for publication. 

Author Response

Thank you very much for your comment. We have tried to improve the manuscript and take into account all the reviewer's suggestions.